# Development of a Virtual Force Sensor for a Low-Cost Collaborative Robot and Applications to Safety Control

**DOI:** 10.3390/s19112603

**Published:** 2019-06-07

**Authors:** Shih-Hsiang Yen, Pei-Chong Tang, Yuan-Chiu Lin, Chyi-Yeu Lin

**Affiliations:** 1Department of Mechanical Engineering, National Taiwan University of Science and Technology, Taipei 106, Taiwan; u060112@hotmail.com; 2Ubiqelife Technology Corporation, Jhubei, Hsinchu 302, Taiwan; tangpc2010@gmail.com (P.-C.T.); stephen@ubiqelife.com (Y.-C.L.); 3Taiwan Building Technology Center, National Taiwan University of Science and Technology, Taipei 106, Taiwan; 4Center for Cyber-Physical System, National Taiwan University of Science and Technology, Taipei 106, Taiwan

**Keywords:** collaborative robot, virtual force sensor, collision detection, safety control, impedance control, stiffness control

## Abstract

To protect operators and conform to safety standards for human–machine interactions, the design of collaborative robot arms often incorporates flexible mechanisms and force sensors to detect and absorb external impact forces. However, this approach increases production costs, making the introduction of such robot arms into low-cost service applications difficult. This study proposes a low-cost, sensorless rigid robot arm design that employs a virtual force sensor and stiffness control to enable the safety collision detection and low-precision force control of robot arms. In this design, when a robot arm is subjected to an external force while in motion, the contact force observer estimates the external torques on each joint according to the motor electric current and calculation errors of the system model, which are then used to estimate the external contact force exerted on the robot arm’s end-effector. Additionally, a torque saturation limiter is added to the servo drive for each axis to enable the real-time adjustment of joint torque output according to the estimated external force, regulation of system stiffness, and achievement of impedance control that can be applied in safety measures and force control. The design this study developed is a departure from the conventional multisensor flexible mechanism approach. Moreover, it is a low-cost and sensorless design that relies on model-based control for stiffness regulation, thereby improving the safety and force control in robot arm applications.

## 1. Introduction

Industrial robot arms in human–machine interaction applications must conform to the safety standards of ISO 15066 [1], which require robot arms to decelerate or fully cease movement when humans are present within their working range. For this reason, the design of collaborative robot arms often includes safety features, such as shock-absorbing series elastic actuators [2] at their joints, or flexible mechanisms, such as the one used by “Baxter” of Rethink Robotics [3], to dampen the force of impact by changing the shape of the joints, thereby reducing the damage incurred during unexpected collisions. Another approach is to install sensors that can detect external contact and supplement them with impedance control to improve safety during human–machine interactions. In the literature, robot arms chiefly detect forces through the following three methods:
(1)Force or torque sensor: This type of sensor is usually installed on an end-effector to measure the contact force on end-of-arm tooling. It is suitable for operations that require high-precision force control on workpieces, such as grinding and polishing [4]. The sensor has the advantage of being highly adaptable and easy to install, but it can only detect the force on the end-of-arm tooling, meaning that it cannot detect possible collisions. Few studies have installed Force/Torque (F/T) sensors on the socket of a robot arm to enable the real-time detection of impulse force on an unspecified location to maintain safety in the case of collision [5]; the FANUC CR-35iA [6] is one example.(2)Joint torque sensor: This type of sensor is installed on joints and usually calculates torque according to the deformation of moveable components in a joint; the KUKA LWR uses such a sensor [7]. Its advantage is that it can detect the force applied on each joint; however, it complicates joint design and increases costs.(3)Virtual force sensor: This type of sensor measures the external torque resulting from an external contact force by calculating the difference between the theoretical torque of the system model in motion and the torque generated by the actual movement of the robot arm. This type of sensor is used in Universal Robots, which perform motor current monitoring for safety control [8], and has the advantages of not requiring additional force sensors and being capable of detecting the torque condition of each joint. However, it is reliant on a high-precision system model and involves complicated kinematic calculations.

F/T sensors are sold at approximately one-tenth of the price of collaborative robot arms (approximately US$ 300). Although this may be an acceptable price for automated industrial applications, it is too expensive for low-cost service applications. Therefore, sensorless force detection methods are more suitable for service applications that do not require high levels of precision.

Studies on virtual force sensors have primarily employed two approaches for internal sensors to detect a force. One is the use of position sensors, which compare differences in position and velocity to estimate the external torque generated by the motor. The other is the use of current sensors, which measure the difference in a controlled current to calculate an external torque. Table 1 presents a comparison of these studies. High precision is not required in force-sensing operations for impedance control or collision detection; thus, torques can be estimated according to the differences in position and velocity without a system model [9]. By contrast, controlling contact force requires precise information of the force, which requires a system model and status information obtained by performing high-precision calculations [10].

Because the control of robot arms requires high precision, it usually employs a high-gain closed-loop system to reduce position errors. This results in high stiffness in the control system, for which reason most collaborative robot arm designs employ flexible joints to absorb external impact forces in a short time and reduce the damage caused by collision. However, flexible joints are a mechanism with a fixed elastic coefficient, which makes them too rigid to adapt to different applications. Moreover, flexible joints are prone to vibration, which affects the precision of control, rendering motor control difficult and increasing the costs of development and production.

In this study, a self-developed low-cost rigid robot arm was employed for force control, and the sensors used were three Hall sensors and a current sensor for each joint. Because low-resolution position sensors provide an insufficient sampling of velocity, thus rendering the calculation of contact forces through high-precision feedback impossible, an open-loop dynamic compensator was added to resolve this problem. For velocity and acceleration, sensor feedback was replaced by motion commands, and dynamic equations were used to calculate the torque required for motion control, allowing direct control of the torque compensation for the motor. After applying torque compensation, the closed-loop controller only needed to correct control errors. This enabled the estimation of an unknown external torque with only current errors. Calculating external contact forces using the system model enabled the use of a small number of sensors for force data.

For safety considerations in human–machine interactions, we proposed a control method that could alter the rigidity of the robot arm by adding a torque saturation limiter to the servo actuator on each axis for the purpose of restraining torque output. This method employs a contact force observer for real-time measurement of contact forces and adjusts the torque saturation to restrain the maximum torque output of each joint, thereby controlling the robot arm’s force output to remain within a safe range. Although this method offers lower precision in force control, it does not involve inverse kinematic calculations or additional revision of the controller structure. Therefore, it effectively reduces the complexity of controller design and processor costs without the need for additional force sensors or flexible mechanisms. For low-cost, rigid service robots and associated applications, this was a solution that can enhance the safety and basic force control of a robot arm without increasing the number of sensors required or hardware costs.

## 2. Model-Based Force Estimation

### 2.1. Contact Force Estimation

The robot kinematic must be considered to detect contact forces without force sensors. In an *n*-joint rigid robot model, assuming joint positions are expressed as q∈ℝn, the coordinates of the link *i* relative to the origin can be expressed as follows:
Tiq=Riqpiq01, i=1,….,n
where Riq∈ℝ3×3 is a rotation matrix and pi∈ℝ3 is a position vector. The velocity of the link *i* can be calculated by applying (1), where Ji∈ℝ6×n is a geometric (or basic) Jacobian, q˙∈ℝn is joint velocity, vi∈ℝ3 is linear velocity, and ωi∈ℝ3 is angular velocity.
(1)viωi=Jiqq˙

In the static state, the contact force generates a different external torque on each joint, so the relative position from a contact point to each joint must be calculated. In Figure 1a, when an external force comes in contact with the surface of link *i*, assuming $p_{c}$ to be the absolute position of the contact point, the relative position of $p_{c}$ to each joint, $p_{i,c}\left(q\right)$, can be expressed as (2). From (1) and (2), the geometric Jacobian matrix of the contact point on link *i*, denoted as $J_{c}$, can be written as (3), where $S\left(v\right)$ is the skew-symmetric matrix of the velocity component. Because the contact force applied on link *i* only affects the joints before link *i*, the final *n − i* columns in the Jacobian matrix $J_{c}$ should be 0.
(2)$$p_{i,c}\left(q\right)=p_{c}\left(q\right)-p_{i}\left(q\right)$$
(3)$$J_{c}\left(q\right)=\left[\begin{array}{cc} I & -S\left(p_{i,c}\left(q\right)\right)\\ O & I \end{array}\right]J_{i}\left(q\right)$$

Assuming the contact force on pc to be Fc∈ℝ3 and the contact torque to be Mc∈ℝ3, the joint external torque τext generated by the contact force can be expressed as (4), which is derived by Luca [12]. This equation enables the calculation of an external contact force from the estimated external torques on the joints without needing to depend on additional force sensors.
(4)τext=JcTq·FcMc

In this study, force sensing was chiefly applied for safety reasons and simple force control. Therefore, only the simple contact force $F_{c}$ was considered, and the external torque $M_{c}$ was disregarded. Moreover, because no additional sensors, such as 3D cameras [12] or touch sensors [15] were employed to detect the point of contact of an external force, training for learning the point of contact of an unknown contact force from position and current sensors was impossible. Therefore, only forces that were applied on the end-effector, as depicted in Figure 1b, were considered. For an external force that was applied on an unknown location, an equivalent was calculated and applied on the end-effector. This equivalent force was, of course, less than the actual force.

### 2.2. External Torque Observer

The dynamic model of an n-joint robot manipulator can be written in the Lagrangian form as [16].
(5)Mqq¨+Cq,q˙q˙+gq+τfq˙=τ+τext
where q∈ℝn is the vector of joint variables, Mq is the inertia matrix, Cq,q˙ is the vector of Coriolis or centrifugal torques, gq is the vector of gravity torques, τfq˙ is the vector of friction torques, τ is the joint control torque, and τext is the vector of external torque due to external generalized contact forces acting on the robot.

Luca et al. [9,12] proposed a method for the estimation of external torques from Equation (5). The residual vector r∈ℝn is defined as.
(6)rt=KIp−∫0tτ+CTq,q˙q˙−gq+rds
where p=Mqq˙ is the generalized momentum of the robot and KI is a diagonal gain matrix. The dynamic evolution of r has the stable, first-order filter structure r˙=KIτext−τ, so that we can assume, for sufficiently large gains, r≃τext. This approach provides a model-based estimate of the external torque resulting from a contact force or torque applied anywhere to the robot by joint position sensors. However, because the low-cost robot arm used in the present study employed only a low-resolution Hall effect sensor as its position sensor, the resolution for the location of the motor was only 24 pulse/rev, resulting in undersampling and phase delay of the velocity data [17]. In addition, the current detection of motor drivers was performed using a single-phase current sensing approach [18], which involved the use of only one current sensor and measuring phase Back Electromotive Force (EMF) to predict the current of the Brushless Direct Current (BLDC) motors. Since the position and current feedback information are not precise, this rendered (6) inapplicable for the calculation of external torques.

To compensate for the low sensor precision, an open-loop controller was introduced. In the dynamic Equation (5), velocity and acceleration commands from a trajectory generator were used for the calculation of torque compensation (7) [19], as shown in Figure 2. After applying the predetermined torque compensation, the closed-loop controller was only responsible for the reduction of control errors, which did not require considerable gain. Under such a control system, the servo controller’s torque output, $\tau_{c}$, could be considered a disturbance torque, which is the sum of the external torque and the error of the compensator’s computed torques (8). In a highly accurate system model, the error of computed torques is small and the disturbance torque can be regarded as the external torque. In this study, low-speed human–machine interactions were largely considered, so the dynamic equation calculation can be simplified or ignored.
(7)$$\tau_{d}=M\left(q\right){\ddot{q}}_{d}+C\left(q,{\dot{q}}_{d}\right){\dot{q}}_{d}+g\left(q\right)+\tau_{f}\left({\dot{q}}_{d}\right)\;$$
(8)$$\tau_{c}=\tau_{\mathrm{error}}+\tau_{\mathrm{ext}}\;$$

### 2.3. External Torque Observer Calibration

When a robot arm is in a static state $\left(\ddot{q}=\dot{q}=0\right)$, its dynamic Equation (5) can be simplified as (9), which only considers gravity, static friction, and external torque. Where gravity compensation is provided, the changes in the motor’s torque can be used as a means to detect external torques on joints. When a motor undergoes a current–torque calibration, a load with a constant mass is used to create a constant torque in the direction of gravity for the purpose of measuring the change in the counter torque generated by the motor. However, because this external torque must be greater than the static friction torque for changes in the motor’s position and current to occur, the static friction torque must be included in the motor’s torque equation during calibration.
(9)$$\tau_{c}=\tau_{g}+\tau_{f}+\tau_{\mathrm{ext}}$$

Friction pertains to a nonlinear system, and its system model can only be obtained through experimentation. A friction identification experiment conducted by Wolf et al. [20] revealed that the friction in joint mechanisms was a complex nonlinear system. The velocity-current experiment proved that at a high velocity (>100 rpm) Coulomb friction can be considered a constant. If the nonlinear torque changes caused by temperature and load are excluded, the friction equation can be simplified as Equation (10), where $F_{c}$ is Coulomb friction, $F_{v}$ is the linear viscous coefficient, and $\dot{\theta }$ is the motor velocity. The friction coefficients can be obtained from the velocity-current experiment. In Figure 3, the two friction coefficients $F_{c}$ and $F_{v}$ could be obtained from the regression line as the slope and the *y*-intercept, respectively. In the static state ($\dot{\theta }≅0)$, static friction torque is equal to $F_{c}$.
(10)$$\tau_{f}=F_{c}sign\left(\dot{\theta }\right)+F_{v}\dot{\theta }$$

Because of differences in manufacturing and assembly conditions, joint mechanisms can generate different levels of friction. For this reason, the current–torque calibration was conducted separately for each axis in this study. By substituting the static friction torque $\tau_{f}$, gravity torque $\tau_{g}$, and external torque $\tau_{ext}$ into Equation (9), the actual torque output of a joint, $\tau_{c}$ could be obtained. Taking the calibration of Joint 2, for example, a posture in which only Joint 2 was subjected to a force along the Z-axis (gravity direction) was chosen. The locations of the robot arm’s joints were as follows:
q=[0, π/2, −π/2, 0, 0, 0, 0]

When the robot arm was in the posture depicted in Figure 4, an external load of 5 kg was attached to the end-effector to induce force along the Z-axis. By adjusting the corrected gain $k_{c}$, the calculated external force was equalized to the applied load, as shown in Figure 5.

### 2.4. Verification of Contact Force Estimation

Current–torque calibration enabled the external torque observer to estimate the external torque on a joint and then calculate the corresponding external contact force by using Equation (4). To verify the accuracy of force estimation, a 6-axis F/T sensor, Robotiq FT300, was installed on the end-effector, as depicted in Figure 6. Specifications of the sensor are listed in Table 2 [21].

In this experiment, a posture that is common among working robot arms was selected. The position of the end-effector based on the robot’s coordinate system was as follows:
P=[x, y, z, rx, ry, rz]=[−0.4 m, 0.0 m, 0.3 m, 180°]

Different force values were applied on the end-effector from the Y, X, and Z axes, and the results of the contact force observer and the F/T sensor were compared, as demonstrated in Figure 7.

The comparison revealed that the contact force observer had a 10% mean error along the X-axis and Y-axis and an approximate mean error of 15% along the Z-axis. Because the estimated external torque included the error of controlled torque compensation, it had a greater error along the Z-axis, which experienced greater changes in gravitational force. Because of friction within the joint mechanism, the contact force observer could not detect an external torque that was less than the static friction torque. Therefore, an external torque of approximately 10 Nm or lower could not be accurately estimated. Additionally, because the dynamic model had been simplified for greater computational efficiency and reaction speed, the accuracy of the contact force observer, which was based on the dynamic model, could be affected by the precision and computational accuracy of the system model. This is a disadvantage of the force estimation method. However, if it is only used in the detection of contact forces for safety and in applications that require only low-precision force control, then a force estimation range of 20–80 N should be sufficient.

## 3. Safety Force Controller for Collaborative Robot

Robot arms that require high-precision control must employ a high-stiffness control loop. The impedance control of contact forces coming from unspecified locations requires the use of additional force sensors to measure a contact force, thus changing the robot arm’s target position and movement trajectory and achieving force control (e.g., Cartesian impedance control [17] or admittance control [22]). Because this method requires real-time inverse kinematic calculations for the displacement of each joint, the controller used must be capable of advanced computation and real-time communication. Figure 8 presents the control diagram of this method.

In the design of a general purpose motor drive, torque saturation is used in the calculation of final torque output, as shown in Figure 8. However, in the applications of a robot arm, the motor must maintain a torque that is sufficient to resist gravity at all times. Any attempt to lower torque saturation or turn off torque output can cause the robot arm to lose power and drop, which is dangerous. For this reason, the emergency stop procedure generally requires a brake to be activated as soon as the motor’s power is cut in order to lock the robot arm in its current position and prevent it from falling on something or someone. Nevertheless, an emergency lock-up can cause the robot arm to pin a person or workpiece in place, resulting in unwanted damage. Therefore, this stopping procedure is not exactly safe. This section presents a safe torque control method to resolve this problem.

### 3.1. Torque Saturation Design

To realize safe robot arm control, the torque saturation limiter in the motor drive is moved in front of the compensator, thereby limiting only the output of the servo controller (Figure 9). In this position, the stiffness of the control system can be changed by tuning the torque saturation without needing to alter the parameters of the closed-loop controller.

During an emergency, the controller can issue a command to cease motion control. Additionally, it can reduce or cease the servo motor’s torque output in a timely manner. With limited torque from the motor, the dynamic compensator can generate a gravity torque, thus maintaining the balance of the robot arm and rendering brake activation unnecessary. Moreover, because the output of the closed-loop controller is restrained, the robot arm can easily be moved by an external force, making it possible to immediately move the robot arm away from a danger zone. This effectively prevents collisions that can lead to injury or damage.

### 3.2. Impedance and Force Control

Robot arm control can enhance safety by reducing controller gain from a closed-loop system, such as through the application of joint impedance control. The joint control torque can be expressed as Equation (11), where $K_{p}$ is the position control gain and $K_{v}$ is the velocity control gain. However, reducing controller gain also reduces the precision of position control. When an application requires different levels of precision and stiffness, changing between different control settings is difficult; moreover, the gain of each joint must be adjusted repeatedly.
(11)$$\tau_{c}=K_{p}\left(q-q_{d}\right)+K_{v}\left(\dot{q}-{\dot{q}}_{d}\right)$$

We hereby present a joint impedance control method that only changes torque output by regulating torque saturation and does not involve the reduction of controller gain. It achieves impedance control without needing to engage in time-consuming parameter adjustment, as is the case for the conventional method. A contact force observer detects the magnitude of a force in real time. When the force is excessively strong, it regulates the torque saturation to reduce the torque output. Because this approach does not involve inverse kinematic calculations, it reduces the burden of the controller system. The cycle for force estimation and force control is 20 ms (Figure 10).

Under this framework of stiffness control, torque saturation is one simple parameter that applies to all axes. The control parameter of the torque saturation limiter is defined as the maximum torque control threshold, *mxC*. Therefore, the estimation of a contact force cannot be achieved according to the force’s components along each axis. Instead, it has to be achieved using the scalar of the resultant force, as expressed in Equation (12), where $F_{x},\;F_{y},\;F_{z}$ are estimated forces on the *x*, *y*, *z*-axis, respectively. Specifically, a contact force observer is used to measure the resultant force at work, and then the torque saturation is regulated to limit the torque output of the joints, and thus achieve low-precision contact force control.
(12)$$F_{res}=\sqrt{F_{x}{}^{2}+F_{y}{}^{2}+F_{z}{}^{2}}$$

### 3.3. Safety Collision Detection

In applications that involve human–machine interactions, robot arms share the working space with human workers, rendering them prone to collisions. For this reason, collision detection is a critical safety feature for collaborative robot arms. The *BG/BGIA Risk Assessment Recommendations According to Machinery Directive: Design of Workplaces with Collaborative Robots* advises the safe range of external forces on different regions of the human body [23]. The lowest is on the neck (front), which is 35 N. Table 3 offers a more detailed list of these data.

Therefore, the design of the safety controller adopted 35 N as the safety threshold for contact forces. When the contact force observer determines that a contact force is greater than this threshold, the controller reduces *mxC* to lower the torque output in joints, thereby decreasing the contact force of a robot arm. Because the contact force observer can only estimate a resultant force that is equivalent to the force on the end-effector, to prevent contact force on an unspecified location from generating an excessively large torque, an adjustable safety threshold is introduced to the external torque observer on each axis to reduce *mxC* when control torque exceeds this threshold. When the cumulative joint position error of a robot arm in motion, which is generated by reduced stiffness, exceeds a predetermined following error, the controller triggers an alarm that stops motion control and switches off the torque output. At this moment, only the gravitational torque compensation of the robot arm is in a state of zero gravity, facilitating the process of swinging the robot arm away to reduce the damage caused by collision. The operation cycle of the controller is set at 20 ms. Figure 11 details the logical architecture of the controller.

## 4. Experiment and Results

In the experiments, this study used a self-developed seven-axis robot arm, which exhibited concise design with low-functionality components to conduct experiments to verify the validity of this study. The specification of the robot arm is shown in Table 4. To reduce the size of controller, this study used embedded drivers designed from a development board of Field-Programmable Gate Array (FPGA) and used open-source software to design real-time controllers prioritizing low cost and employed the Raspberry Pi 3 Module B and Linux Ubuntu MATE system accompanied by a RTOS Xenomai 3 [24]. A Robotiq FT300 sensor was used to collect the data of contact forces on the end-effector to determine the accuracy of force detection and control in this experiment.

### 4.1. Collision Detection and Safety Control

Force detection was divided into two experiments, each with different detection conditions (Figure 12):

(a) Collision detection: The robot arm moved along the Y-axis to Y = −0.22 m at a linear velocity of 100 mm/sec. An object was placed at Y = −0.15 m for collision with the robot arm.

(b) External force detection: The robot arm moved along the Y-axis toward the negative plane at a linear velocity of 100 mm/s, and throughout the duration, an external force along the Y-axis toward the positive plane was exerted on the robot arm.

### 4.2. Experiment 1: Collision Detection

The collision detection experiment consisted of two parts. The first was conducted without safety control, and the results are presented in Figure 13a. The collision occurred at *t* = 1.5 s, and the force of the collision was determined to be approximately 60 N, which exceeds the safety threshold of 35 N. Therefore, the force control was deemed unsafe.

Safety control was introduced in the second part of the experiment and the results are presented in Figure 13b; thus, when the robot arm detected a contact force greater than the safety threshold of 35 N at *t* = 1.6 s, the controller automatically reduced system stiffness to reduce the contact force to within the threshold at *t* = 2.1 s. Comparing two results, the robot arm reduced its system stiffness at collision and completed safety protection measures at approximately 500 ms, reducing the contact force to within the safety threshold. This result demonstrates the usefulness of force detection and stiffness control in collision protection.

### 4.3. Experiment 2: External Force Detection

When a robot arm in motion comes into contact with an unknown external force, it first lowers its stiffness to reduce the contact force to less than the safety threshold. When the motor’s following error exceeds an acceptable range, the controller immediately brings the robot arm to a complete stop and reduces system stiffness to a state in which the torque output becomes zero. 

According to the experimental results presented in Figure 14, the external force was detected at *t* = 1.3 s and the safety controller began reducing stiffness at *t* = 1.6 s. Contact force was detected and controlled within the safety force threshold. At *t* = 2.3 s, an alarm was triggered to fully stop movement of the robot arm when the position error became excessive. At that time, the torque output was reduced to zero, and the motor only generated a gravity torque to keep the robot arm in a state of zero gravity to facilitate its movement by an external force. With these experimental results, it is proved that the stiffness control and emergency stop strategy are real-time and safe.

### 4.4. Force Control Application

The force control experiment tested wiping. In an application that involves wiping a whiteboard, the robot arm needs to bring the tool (eraser) into contact with the working plane (whiteboard) and maintain a contact force for the tool to move along the working surface. Therefore, the controller must be capable of measuring and controlling the contact force in real time. In the experiment, the force control procedure consisted of four phases, Figure 15a–d, as depicted in Figure 15. According to the direction of the controlled contact force, the experiment was further divided into two sub-experiments.

### 4.5. Experiment 3: Contact Force Control

The experimental phase (b) involved the control of a contact force in a single direction. The amount of contact force was set at 30 N and 50 N, the working plane was at Z = 0.25 m, and the robot arm was given a command to move to Z = 0.21 m, which is lower than the working plane. The resultant position error created a force that was controlled to remain along the Z-axis in (11), and the contact force observer was employed to adjust the system stiffness and generate a contact force on the working plane.

According to the experimental results displayed in Figure 16, the robot arm came into contact with the working plane at *t* =1.2 s, generating a rising contact force; the contact force reached 60 N at *t* = 1.4 s, prompting force control to reduce system stiffness and the contact force to the desired level; and the contact force converged to the target level at *t* = 2.2 s. During the stiffness control (*t* = 1.4 ~ 2.2 s), the force estimation error is larger because the motor torque was being controlled in high-frequency. These results suggest that force control was able to bring a unidirectional contact force to a desired level in approximately 600 ms.

### 4.6. Experiment 4: Wiping Motion Control

In the wiping motion in the experimental phase (c), the contact force consisted of a downward component (Z-axis) and a component that caused movement (Y-axis). In this experiment, the robot arm came into contact with the working plane and generated a contact force of 30 N, after which it moved toward Y = −0.15 m at a velocity of 100 mm/s. While in motion, the robot arm maintained a contact force of 30 N against the working plane until finally moving upward and leaving the working plane.

According to the experimental results displayed in Figure 17, the end-effector came into contact with the working plane at *t* = 0.9 s, thus generating a contact force. Moreover, the contact force reached the desired level of 30 N at *t* = 1.7 s; the end-effector started moving along the working plane at *t* = 2.6 s, and the contact force was maintained at 30 N. At *t* = 4.0 s, the robot arm completed its task and moved off of the working plane. Figure 18 shows the positions that the robot arm was commanded to reach and the actual positions that it reached.

In phase (b) and (c), the desired position error of Z-axis generated a contact force on Z-axis in Equation (11). A delay occurred because the contact force was caused by position errors in the force control. Furthermore, because force control involving multiple axes resulted in greater errors in torque calculation, the actual contact force detected by the contact force observer was approximately 15 N on Z-axis, which is lower than the estimated value. On X-axis and Y-axis, force estimations also have a few errors in dynamic motion, especially in phase (c). The results show that the force estimation observer is not accurate in dynamic motion because motors are in high-frequency current control. This accuracy problem of dynamic force estimation can be solved by improving dynamic torque compensation on the high-accuracy dynamic model.

In these experiments, a commercial high-precision force sensor was employed to record the contact force on the end-effector in the collision detection, external force detection, and contact force control sub-experiments. The results proved that the contact force observer and force control could effectively be applied in stiffness control, thereby restricting the output of the robot arm and the contact force as soon as collision occurred. The contact force observer exhibited more precise force control in a static state, with a mean error of approximately 10% or less. However, it exhibited greater error in the multiple-axis force control in a dynamic state. This can be an area for future improvement.

## 5. Conclusions

This study applied a low-cost rigid robot arm system designed for service applications to a situation in which few low-resolution sensors were available and proposed a design that combines a contact force observer and a safety force controller. The design relied on the data from the low-resolution position and current sensors to control torques by an open-loop dynamic compensator, which enabled the system to perform real-time estimation of external torques, as well as estimate the contact force on the end-effector with the system model without needing to employ additional force sensors. Regarding safety control, the design introduced a torque saturation limiter to the servo controller for altering joint torque output on the basis of real-time measurements of contact force; the design also enables adjustment of the robot arm’s system stiffness by altering joint torque output, thus realizing joint impedance control and improving the safety of human–machine interactions. This approach has the advantage of not involving an excessive number of parameters in the adjustment of joint torque output. It also effectively reduces the burden of the processor by not requiring complicated inverse kinematic calculations. Overall, this study successfully developed a low-cost safety controller that is capable of detecting a contact force in real time and changing the stiffness and force output of a robot arm according to its application, thereby improving its safety when used in human–machine interactions. In the future, this approach can be applied to low-cost collaborative robots for use in-home or stores in service applications, and can provide contact force detection and safety position and force control without extra sensors and mechanisms.

## Figures and Tables

**Figure 1 sensors-19-02603-f001:**
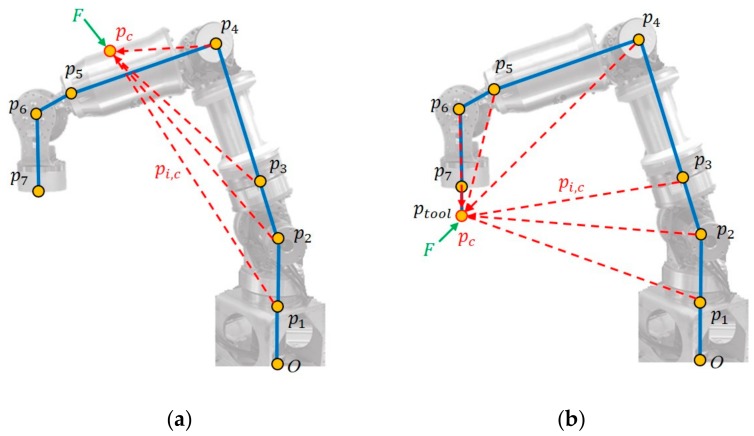
(**a**) Contact force on link 4; (**b**) Contact force on end-effector.

**Figure 2 sensors-19-02603-f002:**
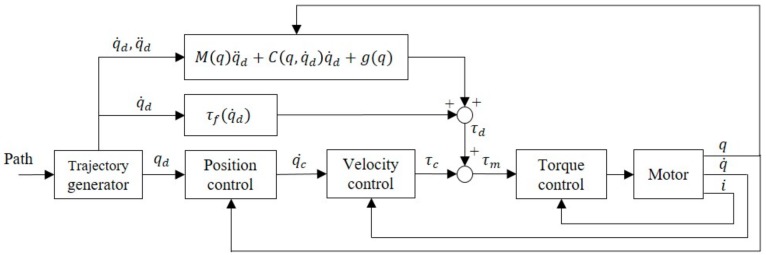
Dynamic compensation control diagram.

**Figure 3 sensors-19-02603-f003:**
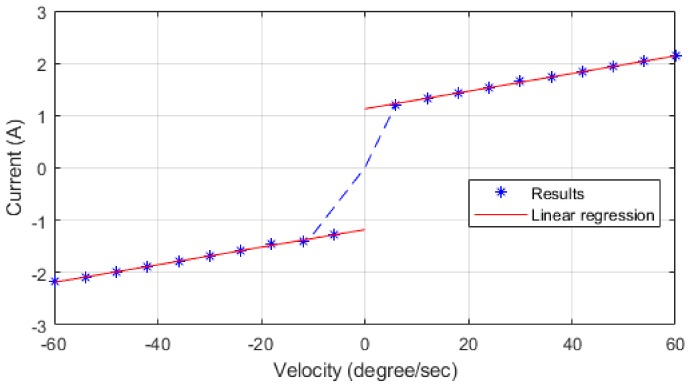
Velocity–current diagram.

**Figure 4 sensors-19-02603-f004:**
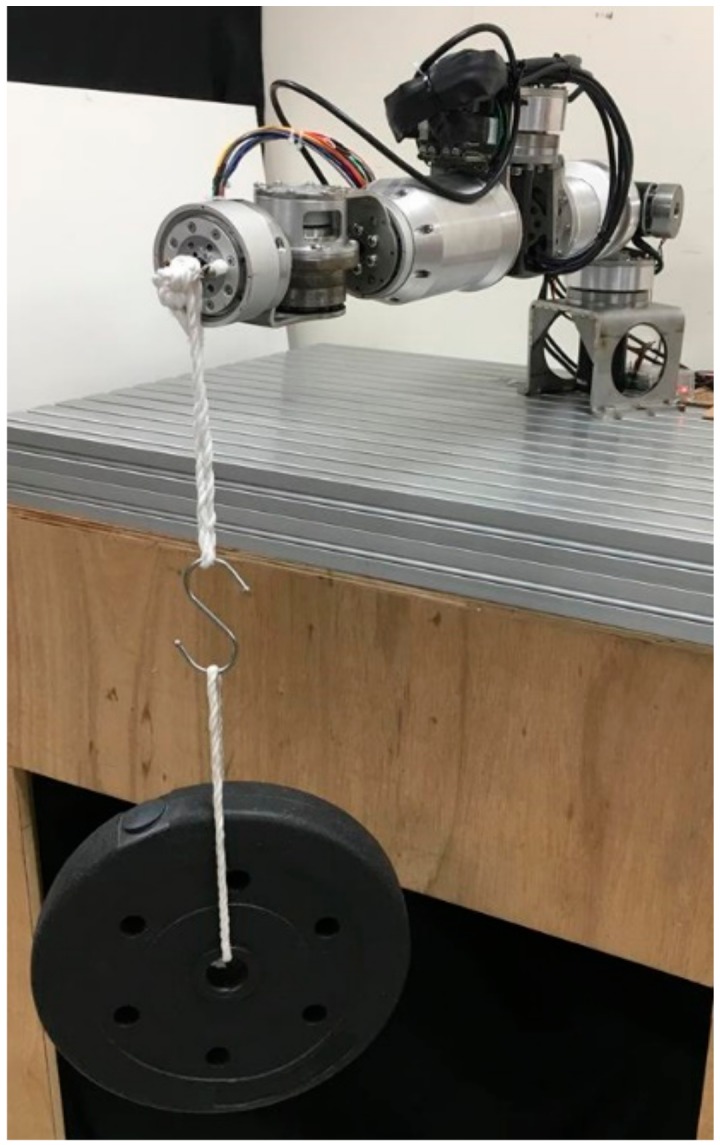
Calibration of Joint 2.

**Figure 5 sensors-19-02603-f005:**
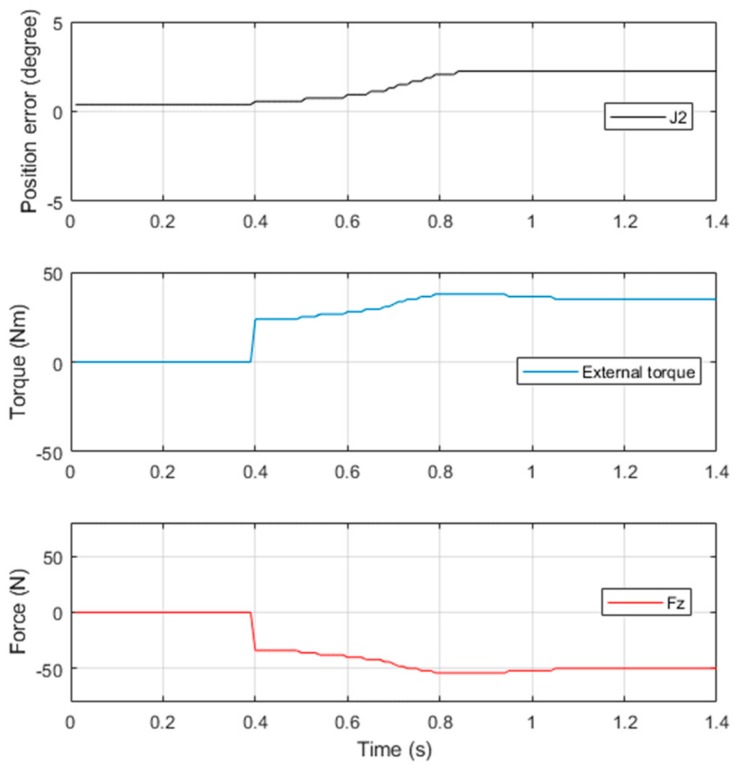
Calibration result of Joint 2 by 5 kg load.

**Figure 6 sensors-19-02603-f006:**
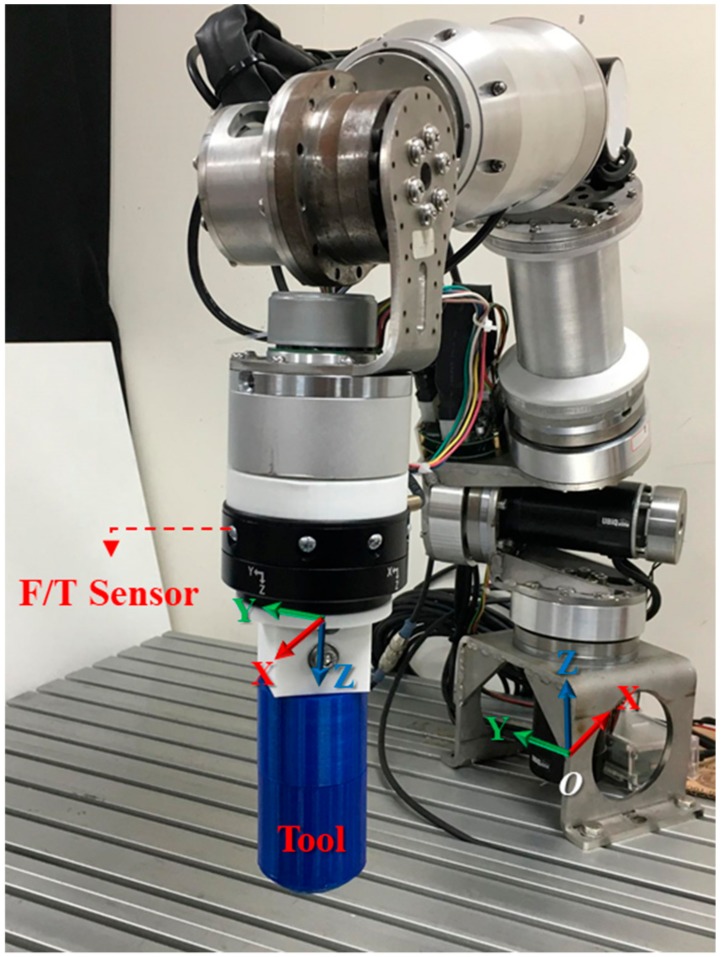
Coordinate system on Force/Torque sensor.

**Figure 7 sensors-19-02603-f007:**
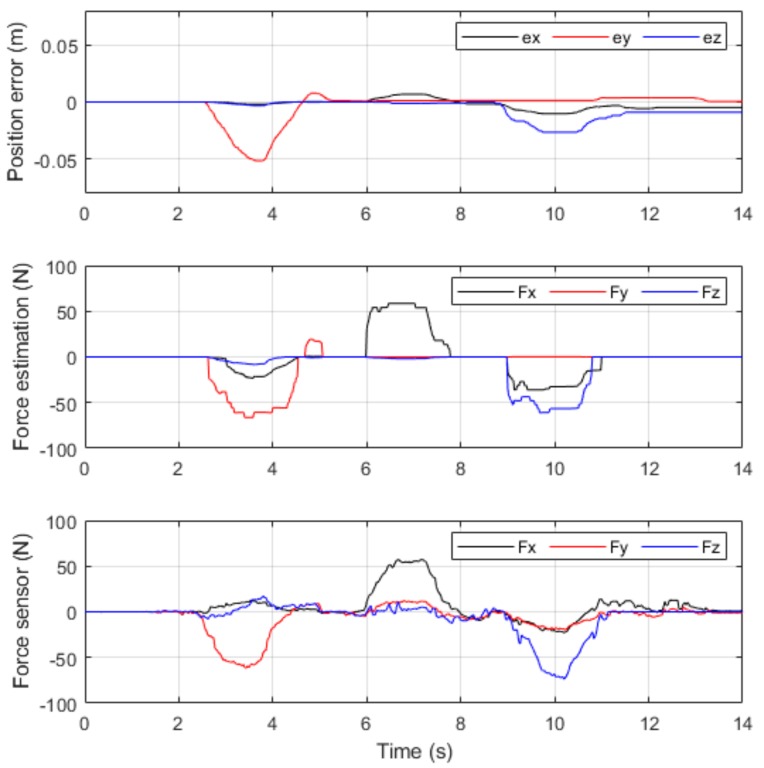
Contact force detection results.

**Figure 8 sensors-19-02603-f008:**
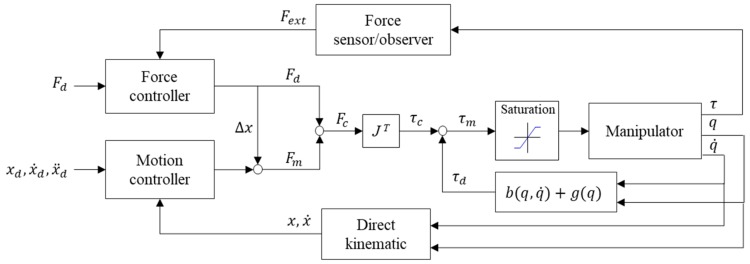
General force control diagram.

**Figure 9 sensors-19-02603-f009:**
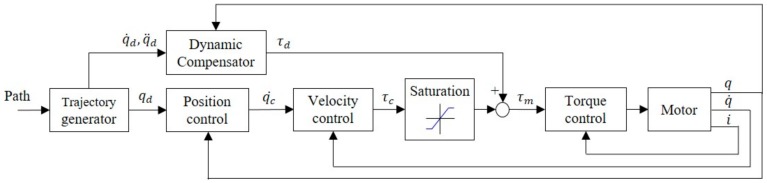
Real-time stiffness control diagram.

**Figure 10 sensors-19-02603-f010:**
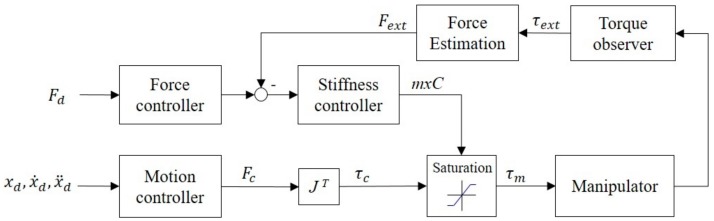
Real-time stiffness control diagram.

**Figure 11 sensors-19-02603-f011:**
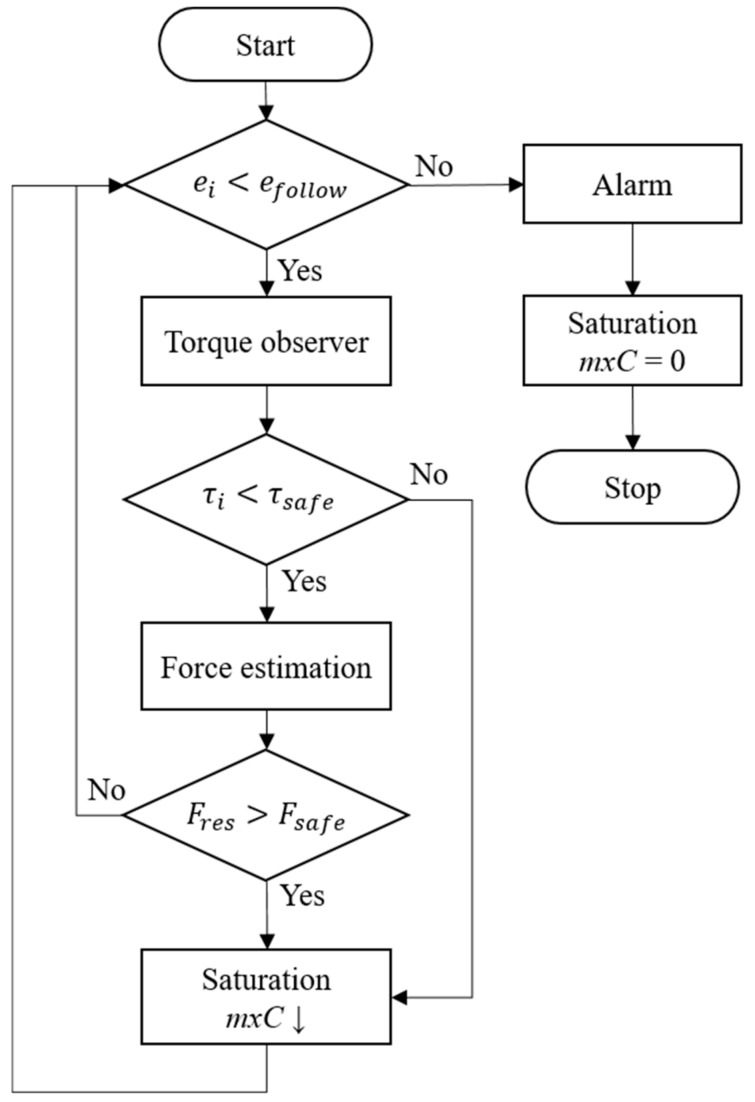
Logical architecture of the force detection and control operations for safety.

**Figure 12 sensors-19-02603-f012:**
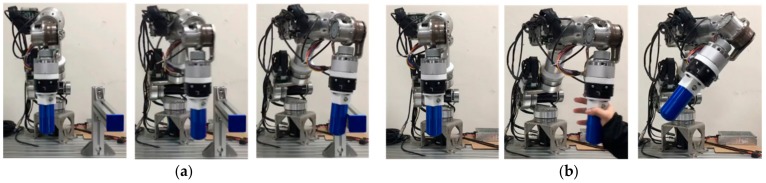
Safety control experiment: (**a**) Collision detection; (**b**) External force detection.

**Figure 13 sensors-19-02603-f013:**
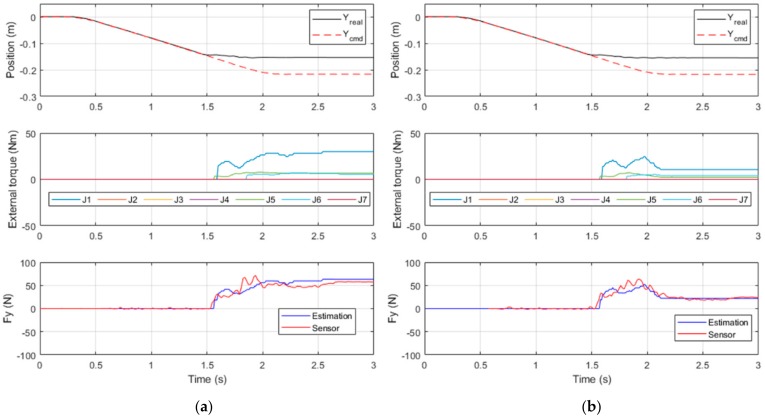
Results of collision detection without safety control (**a**) and with safety control (**b**).

**Figure 14 sensors-19-02603-f014:**
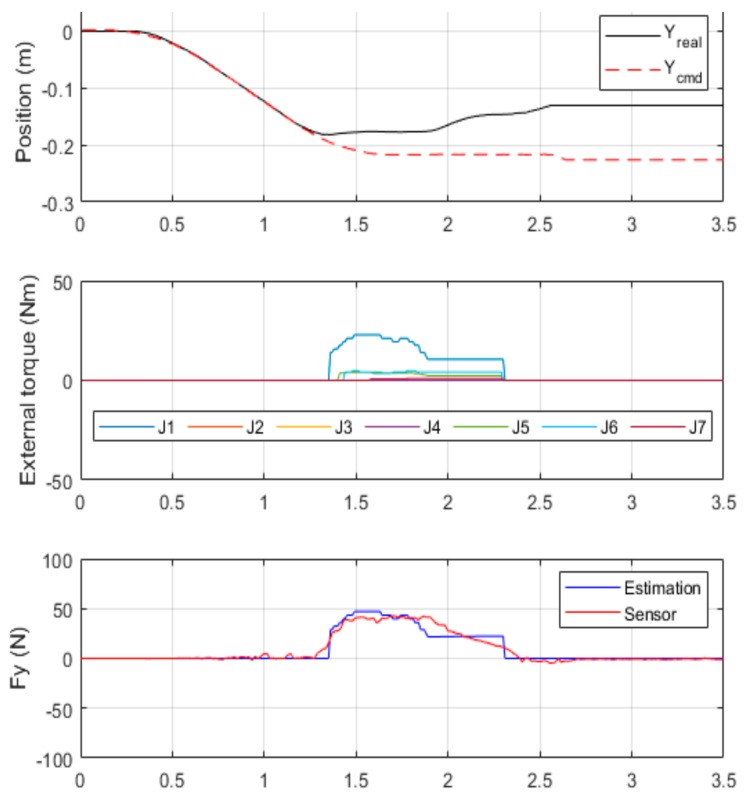
Results of external force detection with a safety controller.

**Figure 15 sensors-19-02603-f015:**
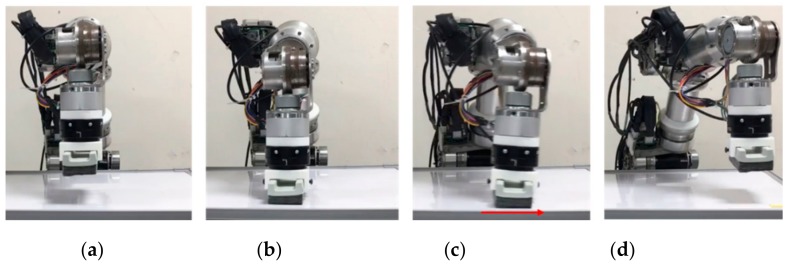
(**a**) Non-contact situation; (**b**) contact situation; (**c**) contact and force application; (**d**) and contact loss.

**Figure 16 sensors-19-02603-f016:**
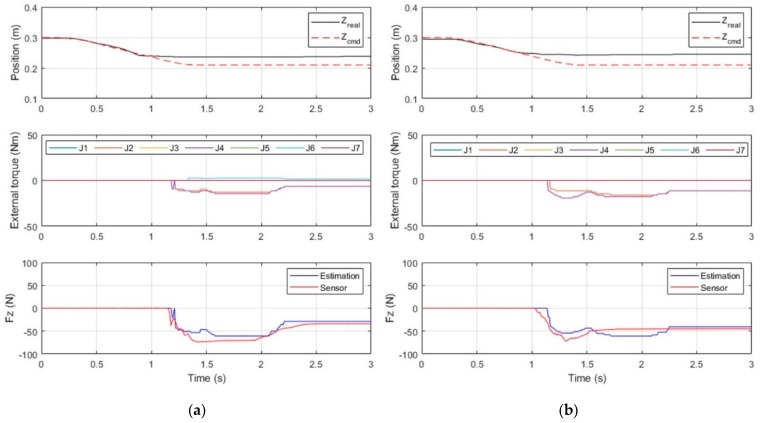
Results of the contact force control experiment: desired forces of 30 N (**a**) and 50 N (**b**).

**Figure 17 sensors-19-02603-f017:**
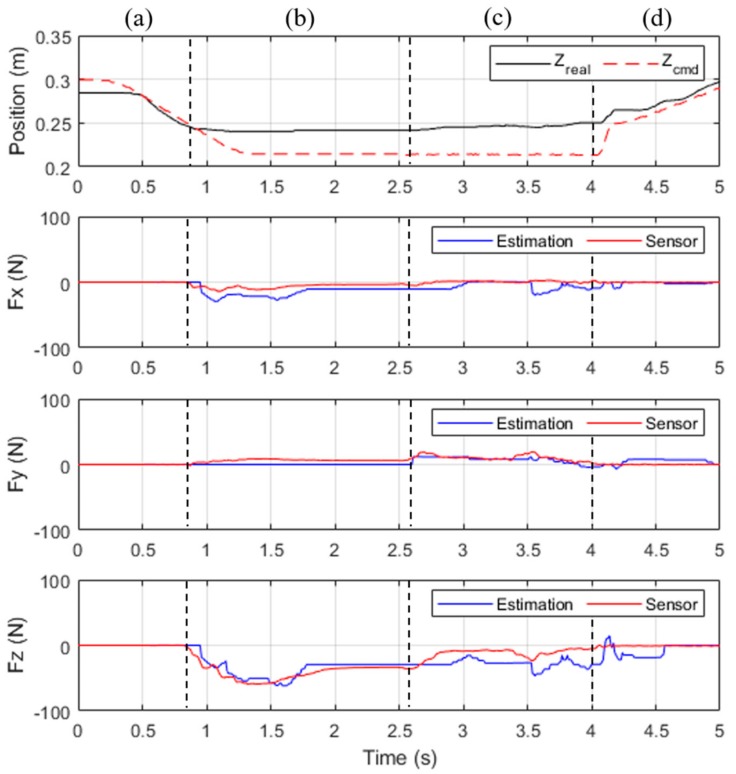
Results of wiping experiment.

**Figure 18 sensors-19-02603-f018:**
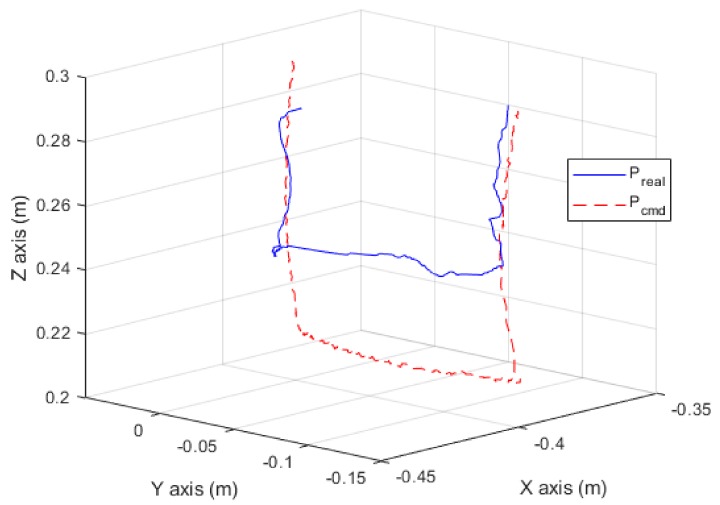
Trajectory of the robot arm in the wiping experiment.

**Table 1 sensors-19-02603-t001:** Comparison of virtual force sensor research.

Reference	[11]	[12]	[13]	[14]	Proposed Approach
Robot	KUKA KR500	KUKA LWR4	HHI HA020W	ABB YuMi	Self-developed
D.O.F.	6	7	6	7	7
Joint	Flexible	Flexible	Flexible	Flexible	Rigid
Sensors	Position Current	Position 3D camera	Position	Position Current	Position Current
Position of force detection	End-effector	Each link	End-effector	End-effector	End-effector
Application	Force control	Impedance and Force control	Impedance and Force control	Force control	Impedance and Force control

**Table 2 sensors-19-02603-t002:** Robotiq force torque sensor FT300 specifications.

Specification	Force	Moment	
	XYZ axis	XY axis	Z axis
Measuring range	±300 N	±30 Nm	
Signal noise	0.1 N	0.05 Nm	0.003 Nm
Recommended threshold	1 N	0.02 Nm	0.01 Nm
Tool deflection	0.01 mm	0.17 deg	0.09 deg
Data output rate	100 Hz		
Communication protocol	Modbus RTU and Data stream (RS-485)		
Mass	300 g		

**Table 3 sensors-19-02603-t003:** Limit values for the forces, pressure, and body deformation constant according to the body regions.

Body Regions	Skull	Face	Neck (Sides)	Neck (Front)	Shoulder	Chest	Arm	Leg
CSF (N)	130	65	145	35	210	140	160	140
IMF (N)	175	90	190	35	250	210	220	170
PSP (N/cm^2^)	30	20	50	10	70	45	50	45
CC (N/mm)	150	75	50	10	35	25	40	60

CSF: Clamping/Squeezing force; IMF: Impact force; PSP: Pressure/Surface pressing; CC: Compression constant.

**Table 4 sensors-19-02603-t004:** The specification of self-developed robot arm.

Specification	Value
Number of axes	7
Weight	15 kg
Payload	5 kg
Reach	700 mm
Repeatability	±2 mm
Power	24 V–10 A
Cost	5000 USD

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
