# Peer review of "Development of a Virtual Force Sensor for a Low-Cost Collaborative Robot and Applications to Safety Control"

_sensors, 2019, doi:10.3390/s19112603_

Round 1

Reviewer 1 Report

This paper proposes a low-cost force estimation method for robot manipulators using motor current information. Controller design methods for tasks under environmental contact were also proposed.

Various application results are presented, where the proposed method is compared with conventional force sensor method, to evaluate the performance of the proposed method. 

I think the following discussion contains unclear statements. Please consider improvement.

1)

Eq.(4) (the relation between the wrench of the hand and joint torques) 

As the formula (4) is the fundamental of the proposed method, clear definitions and descriptions are necessary. Explanations from  physical perspective can help us to avoid misunderstanding. If the formula comes from other paper or textbook please show it in the  reference list. 

In typical textbooks, (for example “A Mathematical Introduction to Robotic Manipulation Richard M. Murray, Zexiang Li, S. Shankar Sastry” ) the idea of equivalent wrenches are defined using equivalence of instantaneous works. If the proposed method is derived from the same premise please state that for clear understanding.

Technical terms and words should be common ones so to prevent misreading. 

For example, “Geometric Jacobian” (in eq.(1)) , seems refer to as “Basic Jacobian”  (A Unified Approach for Motion and Force Control of Robot Manipulators: The Operational Space Formulation, Oussama  Khatib, IEEE JOURNAL OF ROBOTICS AND AUTOMATION, VOL. RA-3, NO. 1, 1987)

2)

If Fig. 2, the signal q[n] seems not well defined. 

3)

Introduction of the corrected gain ( k_c in eq.(10) ) , which seems be necessary to calibrate the motor constant, makes the discussion complicated. 

(I understand it necessary when executing experiment, but is not necessary in discussion.) 

4)

The precise friction model (11) should not be mentioned if it is not used in the proposed method. As the heat affects largely under gravity conditions (as in the experiments in this paper) , we feel difficulty to ignore the effect of temperature change. The simple model (12) is easy to accept even without mentioning the precise model (11).

As seen in the calibration results, proposed method seems not intending a precise method of measuring the force. Hence the precise model of friction can misread us. 

5)

Specification of the motor driver unit, measurement method of the current should be presented. (Measurement of the current is an important in the proposed torque estimator. )

6)

Although the control equations include dynamic inertia forces and non-linear terms, experiments are performed at low speeds where these effects do not appear largely. The validity of the experimental results seems to be limited to the range of statics.  If the experiments are carried out using static model, please mention this point. 

7)

F_x, F_y, F_z in eq.(14) are not defined. 

8)

The estimated results of the hand forces during the wiping experiment are not compared with force sensor results. The comparison result will be a good help for us to understand the applicability to various tasks.

Author Response

Dear Editor:

We are very grateful to comments provided by two reviewers for improving the quality of the manuscript. According to their advice, we carefully amended the manuscript. We sincerely believe that the revisions have fully addressed the concerns provided by reviewers. The reviewers’ comments and our responses are listed below.

To 1st reviewer

Point 1:

Eq.(4) (the relation between the wrench of the hand and joint torques) 

As the formula (4) is the fundamental of the proposed method, clear definitions and descriptions are necessary. Explanations from physical perspective can help us to avoid misunderstanding. If the formula comes from other paper or textbook please show it in the reference list. 

In typical textbooks, (for example “A Mathematical Introduction to Robotic Manipulation Richard M. Murray, Zexiang Li, S. Shankar Sastry” ) the idea of equivalent wrenches are defined using equivalence of instantaneous works. If the proposed method is derived from the same premise please state that for clear understanding.

Technical terms and words should be common ones so to prevent misreading. 

For example, “Geometric Jacobian” (in eq.(1)) , seems refer to as “Basic Jacobian”  (A Unified Approach for Motion and Force Control of Robot Manipulators: The Operational Space Formulation, Oussama  Khatib, IEEE JOURNAL OF ROBOTICS AND AUTOMATION, VOL. RA-3, NO. 1, 1987)

Response 1:

Thank you for the suggestion, we actually missed to cite the source of Eq(4). Eq(1)-(4) come from the conference paper “Estimation of Contact Forces using a Virtual Force Sensor[12]. We have included the reference list and also added a description in the revised article. (P. 3, Line 124-126)

According to textbooks and references, the terminology “Geometric Jacobian” and “Basic Jacobin” were both used. Therefore, I select to use both words. (P. 3, Line 112).

References:

Basic Jacobin:

[1] A Unified Approach for Motion and Force Control of Robot Manipulators: The Operational Space Formulation

https://ieeexplore.ieee.org/stamp/stamp.jsp?tp=&arnumber=1087068

[2] Textbook “Introduction to robotic” Chapter4 p.87

http://robotics.itee.uq.edu.au/~metr4202/2013/tpl/Chapter%204%20-%20Jacobain%20-%20from%20Khatib%20-%20Introduction%20to%20Robotics.pdf

Geometric Jacobian:

[3] Robot Dynamics Lecture Notes

https://www.ethz.ch/content/dam/ethz/special-interest/mavt/robotics-n-intelligent-systems/rsl-dam/documents/RobotDynamics2016/RD2016script.pdf

[4] Geometric Jacobians Derivation and Kinematic Singularity Analysis for Smokie Robot Manipulator & the Barrett WAM

[5] Estimation of Contact Forces using a Virtual Force Sensor

https://ieeexplore.ieee.org/stamp/stamp.jsp?tp=&arnumber=6942848

Point 2:

In Fig. 2, the signal q[n] seems not well defined. 

Response 2:

Thanks for the correction. q[n] means that q is the vector of joint variables on Line 143. We think that it is unnecessary to appear on Figure 2 and Figure 9 so that 2 figures have been replotted. (Figure 2 on P.5 & Figure 9 on P.8)

Point 3:

Introduction of the corrected gain ( k_c in eq.(10) ) , which seems be necessary to calibrate the motor constant, makes the discussion complicated. 

(I understand it necessary when executing experiment, but is not necessary in discussion.) 

Response 3:

We agree that the calibration of motor current constant is not necessary in discussion. The description about the corrected gain (Eq(10) in previous article) has been removed. We only discuss effects of gravity and friction torque on the external torque observer. (P5, Line179-181)

Point 4:

The precise friction model (11) should not be mentioned if it is not used in the proposed method. As the heat affects largely under gravity conditions (as in the experiments in this paper), we feel difficulty to ignore the effect of temperature change. The simple model (12) is easy to accept even without mentioning the precise model (11).

As seen in the calibration results, proposed method seems not intending a precise method of measuring the force. Hence the precise model of friction can misread us.

Response 4:

In this approach, we only use the simple friction model to measure the friction torque. We agree that the precise friction model may cause confusion for readers so we have removed Eq(11) in previous article. (P.5, Line 186-190)

Point 5:

Specification of the motor driver unit, measurement method of the current should be presented. (Measurement of the current is an important in the proposed torque estimator.)

Response 5:

Thanks for the precious suggestion. In a motor driver unit, we used only one current sensor to detect the motor current by the single-phase current sensing approach. We have improved the detail discussion. (P.4, Line 157-160)

Because the current sensor is not precise, the current feedback information is not straightly used to measure the external torque. To solve this problem, we used the servo controller’s torque output, torque_c to measure the external torque. (Discussed on P.5, Line 165-169)

Point 6:

Although the control equations include dynamic inertia forces and non-linear terms, experiments are performed at low speeds where these effects do not appear largely. The validity of the experimental results seems to be limited to the range of statics. If the experiments are carried out using static model, please mention this point.

Response 6:

We agree with your assessment. In this approach, the general equation includes both static and dynamic states. In fact, we consider more on human-machine interactions in low-speed motion, so the dynamic equation calculation can be simplified or ignored. We have revised the discussion accordingly. (P.5, Line 169-170)

Point 7:

F_x, F_y, F_z in eq.(14) are not defined.

Response 7:

Thanks for the correction. Fx, Fy, Fz are the estimated forces on x, y, z-axis. We have rewritten a clear description. (P.9, Line 289)

Point 8:

The estimated results of the hand forces during the wiping experiment are not compared with force sensor results. The comparison result will be a good help for us to understand the applicability to various tasks.

Response 8:

Thanks for the great suggestion. In the wiping experiment, we used a virtual force sensor to detect and control force. A commercial force sensor was only used on recording the actual contact force. We have replotted Figure 17 to show the comparison between the force estimation and the force sensor in each axis. The results show that the force estimation observer is not accurate in dynamic motion because motors are in high-frequency current control. We have revised to a more detailed discussion. (P.14, Line 407-415.)

Reviewer 2 Report

This paper presents a novel observation approach to improve safety features of collaborative robotics and to help with force control.  Overall this paper is strong in writing, structure, and experimental design.  I recommend its acceptance.

There are a few minor typos that I caught.  There may be more, but one more passthrough edit should be sufficient.

Line 85:  I believe it should read: "three Hall effect sensors and a current sensor for each joint"

Line 95: "adding a torque saturation limiter"

Some additional comments to for the review:

While this paper is interesting and well written, here are a few comments/questions that may be addressed to strengthen the manuscript.  I very much like the approach of using relatively low cost sensors and strategies for low precision robotics.

1.  In experiment 1, you show results but don't really interpret them for the reader.  Is the reader just supposed to know what everything in the figures is?

2. In experiment 2, the text describing the results in the figure is too short, meaning that I am not clear on what I am looking at.

3. Looks like experiment 3 that the estimator performed fairly well but there is still error between the estimator and sensor.  Could you explain this and describe what can reduce the error?

4. Looking at the results and description of the experiment shown in Figure 17, what is the cause of the position error?  How can you minimize this?  What changes may be made to the control design to accommodate this?  Why is the actual force lower than the estimated?  Make it clear if this is good or bad.

5. In your conclusion, you talk about how you succeeded in your goal.  However, you do not make clear what avenues of future work you might have on this project.  Could you please expand on this?

Author Response

Dear Editor:

We are very grateful to comments provided by two reviewers for improving the quality of the manuscript. According to their advice, we carefully amended the manuscript. We sincerely believe that the revisions have fully addressed the concerns provided by reviewers. The reviewers’ comments and our responses are listed below.

To 2nd reviewer

Point 1:

In experiment 1, you show results but don't really interpret them for the reader.  Is the reader just supposed to know what everything in the figures is?

Response 1:

Thanks for the comment. In experimet1, we try to explain the effect of the safety stiffness control by the same motion in two sub-experiments. According to Figure 13, the result without the safety control (left-side) shows that unsafety contact force occurred, and the result with the safety control (right-side) shows that the contact force is reduced within safety threshold.

We have written a clear discussion. (P.11, Line 346-349)

Point 2:

In experiment 2, the text describing the results in the figure is too short, meaning that I am not clear on what I am looking at.

Response 2:

In experiment 2, we want to show that the stiffness controller can reduce the contact force (such as experiment 1) and the emergency stop in safety when the over-power contact force occurred in motion. Safety stop strategy with gravity compensation is the main content of discussion in this experiment.

Again, thanks for the comment. We have rewritten a clearer conclusion for experiment 2. (P.12, Line:360-363).

Point 3:

Looks like experiment 3 that the estimator performed fairly well but there is still error between the estimator and sensor. Could you explain this and describe what can reduce the error?

Response 3:

In this approach, we think the dynamic force estimation error is hard to reduce because the motor current must be controlled and detected at the same time. In high-frequency dynamic control, the current detection is unstable.

We have rewritten the discussion on error analysis. (P.13, Line 388-389) The solution of the dynamic problem was discussed at the end of experiment 4. (P.14, Line 412-415)

Point 4:

Looking at the results and description of the experiment shown in Figure 17, what is the cause of the position error?  How can you minimize this?  What changes may be made to the control design to accommodate this?  Why is the actual force lower than the estimated?  Make it clear if this is good or bad.

Response 4:

You have raised an important question. In Figure 17, the Z-axis position error is what we expected. To create a Z-axis contact force in the wiping application, the over-plane position was desired (P.13, Line 380-384). The control force is generated from Eq(11) by the position error.

On the discussion of the estimation errors, we have replotted Figure 17 to show comparison results with the force estimation and the force sensor in each axis. The results show that the force estimation observer is not accurate in dynamic motion because motors are in high-frequency current control. This accuracy problem of the dynamic force estimation can be solved by improving the dynamic torque compensation on the high-accuracy dynamic model.

We have revised this discussion on Line 406-414, P.14, accordingly.

Point 5:

In your conclusion, you talk about how you succeeded in your goal. However, you do not make clear what avenues of future work you might have on this project. Could you please expand on this?

Response 5:

Thank you for your suggestion. In future work, we hope the low-cost safety controller can be used on collaborative robots for in-home or stores, and the robots can conduct easy service tasks such as cleaning, wiping, or picking up/placing down object without extra sensors and mechanisms.

Accordingly, we have revised the discussion of future work on Line 442-444, P.15.

Round 2

Reviewer 1 Report

Please clarify the meaning of the items in the table 1.

Is the term “Approach” represents the method in this paper ? 

Is the term “Detect force” represents the position of the force sensor ?

What does the term “rigid” represents ?

I still don’t understand why eq.(4) is used instead of usual formula between wrench and torque via basic Jacobian matrix. ( why the vectors p_{i,c} are introduced ) 

Please check if they used p_{i,c} vectors in order to utilize 3D cameras ( in the cases of [9,12] ). 

Author Response

Dear Editor:

We are very grateful to comments provided by the reviewer for giving more chances of improving the quality of the manuscript. According to advice, we carefully amended the manuscript. We sincerely believe that the revisions have fully addressed the concerns provided by the reviewer. The reviewer’ comments and our responses are listed below.
